# The Observation of Ligand-Binding-Relevant Open States of Fatty Acid Binding Protein by Molecular Dynamics Simulations and a Markov State Model

**DOI:** 10.3390/ijms20143476

**Published:** 2019-07-15

**Authors:** Yue Guo, Mojie Duan, Minghui Yang

**Affiliations:** 1Key Laboratory of magnetic Resonance in Biological Systems, State Key Laboratory of Magnetic Resonance and Atomic and Molecular Physics, National Center for Magnetic Resonance in Wuhan, Wuhan Institute of Physics and Mathematics, Chinese Academy of Sciences, Wuhan 430071, China; 2University of Chinese Academy of Sciences, Beijing 100049, China; 3Wuhan National Laboratory for Optoelectronics, Huazhong University of Science and Technology, Wuhan 430074, China

**Keywords:** fatty acid binding protein, invisible intermediate states, transition time, binding mechanism, molecular dynamics simulation

## Abstract

As a member of the fatty acids transporter family, the heart fatty acid binding proteins (HFABPs) are responsible for many important biological activities. The binding mechanism of fatty acid with FABP is critical to the understanding of FABP functions. The uncovering of binding-relevant intermediate states and interactions would greatly increase our knowledge of the binding process. In this work, all-atom molecular dynamics (MD) simulations were performed to characterize the structural properties of nativelike intermediate states. Based on multiple 6 μs MD simulations and Markov state model (MSM) analysis, several “open” intermediate states were observed. The transition rates between these states and the native closed state are in good agreement with the experimental measurements, which indicates that these intermediate states are binding relevant. As a common property in the open states, the partially unfolded α2 helix generates a larger portal and provides the driving force to facilitate ligand binding. On the other side, there are two kinds of open states for the ligand-binding HFABP: one has the partially unfolded α2 helix, and the other has the looser β-barrel with disjointing βD-βE strands. Our results provide atomic-level descriptions of the binding-relevant intermediate states and could improve our understanding of the binding mechanism.

## 1. Introduction

Intracellular lipid binding proteins (iLBPs) are critical to the transportation of lipid ligands between tissues and within cells [1,2,3]. As a member of the iLBP super-family, heart-type fatty acid binding protein (HFABP) is abundant in the heart and other tissues like skeletal muscle and brain, and it mainly serves as the transporter of fatty acids from the cell membrane to the mitochondria [4]. The binding and transporting of fatty acids with HFABP play important roles in the metabolism signaling pathway, and abnormal function of HFABP can induce many serious diseases, such as coronary heart disease, myocardial injury, and various neurodegenerative diseases [5].

Similar to the other FABPs, the fatty acid binding pocket of HFABP has an inner β-barrel, composed of ten antiparallel β-strands (βA-βJ), capped with two short α-helices (α1 and α2) (Figure 1) [6]. The structures of ligand-binding HFABP (holo-HFABP) obtained in both crystal and solution states are nearly identical to those of the ligand-free HFABP (apo-HFABP). It is believed that the α1 helix and α2 helix, along with the βC-βD and βE-βF loops, comprise the dynamic portal which allows the entrance and existence of fatty acids in the inner binding pocket of FABP [7,8,9]. However, the detailed ligand binding mechanisms of FABP remain poorly understood.

Two hypothetic models have been proposed to describe the binding process. The “open-and-close” model supposes that the helical domain and the C-D and E-F loops form a ligand access gate [10,11,12]. When the helices keep away from the loops, the binding gate is open and allows ligands to enter or escape. Nevertheless, the “open-and-close” model has been challenged by the lack of observation of an open state, with only the “closed” structures characterized by experiments [13,14]. In addition, by introducing a disulfide bond formed by two mutated cysteines which were located in the α2 (K27C) and E-F β turn (D74C) of human intestinal FABP, Yang et al. studied the FA binding ability of gate-blocked FABP. Although the binding affinity was decreased by 10 times, the FABP variant still displayed uptake of fatty acids [15]. Based on this observation, as well as the unfolding of the α2 helix revealed by hydrogen–deuterium (H–D) exchange experiments, they proposed the “local unfolding” model, in which the α2 helix on the FABP would partially unfold and form an open gate for ligand uptake and release [15,16].

The characterization of the intermediates is therefore important to understanding the binding mechanism of ligands to FABP. Relaxation NMR studies have indicated that there are nativelike minor states which play a critical role in the process of ligand binding and unbinding [15]. The time scale of ligand-binding-relevant conformational exchanges in the portal region of human liver FABP is microseconds or even sub-microseconds at room temperature [17,18]. The binding-relevant intermediates, which are functionally important in the ligand binding process and are defined by the experimentally determined ligand association rate in the native closed state [15], are “invisible” and difficult to determine by experimental technologies. For instance, although several minor states of human intestinal fatty acid binding protein were observed by combining ^15^N relaxation dispersion (RD) and chemical exchange saturation transfer (CEST) experiments, Yu and Yang found that these states are irrelevant to the function of fatty acid transport due to the slow conversion rates from the “closed” state to these intermediate states [16].

Molecular dynamics (MD) simulation has been widely used to study the structure dynamics of FABP and the interactions between these proteins and their ligands [12,19,20,21,22,23]. Tsfadia and coworkers successfully observed the entering of a palmitate molecule into toad liver FABP from solution by MD simulation, and they concluded that lipid binding is a rapid process once the FABP is open and the lipid is in the correct orientation [21]. Yang’s group revealed multiple possible dissociation paths by utilizing random expulsion molecular dynamics simulation. The proposed portals were further studied by some short MD simulations at 300 K; the outside palmitic acids (PLM) went through the portal and entered the FABPs in the several-nanosecond simulations [12]. Matsuoka et al. compared the water clusters and H-bond networks inside the binding cavity in apo- and holo-FABP3, and they found that ligand binding induced ordered water clusters and stabilized the H-bond network [22]. Li et al. studied conversions between the open and closed states of FABP4 and found that the ligand-free FABP4 prefers the closed state [23]. Despite the above progress, we still lack full descriptions of the structural properties of the binding-relevant states of FABPs. Furthermore, the conformational transition between the open and closed forms, which is directly related to the ligand binding mechanism, has not been investigated. 

In this study, long unbiased molecular dynamics simulations and a Markov state model (MSM) were employed to study the binding-related intermediate states and kinetic properties of HFABP. Several open intermediate states were characterized. The transition time from these states to the closed native state ranged from a few microseconds to hundreds of microseconds, which is consistent with the experiment estimated transition time. Compared with apo-HFABP, holo-FABP has a larger population of “closed” native conformations, suggesting that ligand binding stabilizes FABP to the closed state. Further, the main structural feature of the intermediate of apo-FABP is the partial unfolding of α2, while the holo-FABP has both α2 unfolding and the separation of βC-βD and βE-βF. 

## 2. Results and Discussion

### 2.1. Structural Properties

The secondary structures including the α1 helix and all β-strands were well preserved in the simulations of both apo-HFABP and holo-HFABP (Appendix A). However, the α2 helix, located in the ligand binding portal, is relatively flexible in both systems. The helicity is lower than 50% for the C-terminal residues on α2 and is about 65% for the overall α2 helix in apo-FABP. Low helicity in the C-terminus of the α2 helix was observed in all the simulated trajectories rather than in a single trajectory, indicating that unfolding of the α2 helix does not appear only occasionally. Ligand binding would increase the stability of the α2 helix. The helicity of residues on the C-terminus of the α2 helix is about 60% and the average helicity is 79% for the overall α2 helix of holo-HFABP (Figure 2A). In fact, the stabilization of secondary structures by ligand binding, especially of the portal region helix, has been observed in experiments on many members of the iLBD super-family [24].

The fraction of native contacts (*Q*) has been commonly used as a reaction coordinate to describe the process of folding [25]. We analyzed the distribution of *Q* for the conformations sampled in our simulations. For both the apo- and holo-FABP systems, the *Q* values are distributed in the range of 0.85 to 0.95 (Appendix A), which indicates that the sampling in our simulation is close to the native state, instead of the unfolding states. The fraction of residue native contacts (*Q_i_*) [26] was calculated to monitor the nativeness of each residue (Figure 2B). The low *Q_i_* values are basically located on the loop regions, such as the βC-βD loop, βE-βF loop, βH-βI loop, and βI-βJ loop, which indicates that these loops are much more flexible than others. Besides this, upon the binding of fatty acid, the *Q_i_* values for the residues on the C-terminal α2 helix increase by about 0.2, demonstrating a population shift of the conformational space toward the folded state induced by ligand binding.

The backbone conformational entropy of each residue is given in Figure 2C. The conformational entropy is related to the change in mechanical nature of the more rigid and more flexible states which describe protein stability and function. Except for the terminal residues, most of the residues with large conformational entropies are located in the ligand entrance portal for both apo- and holo-HFABP. The residues with entropy greater than 2 cal/mol*K are crowded on the C-terminus of the α2 helix and the βC-βD loop, indicating large conformational exchanges in the corresponding regions. The conformational entropies of the residues on the α2 helix of holo-HFABP are smaller than those on apo-HFABP, which further demonstrates the structure stabilization ability of ligand binding. Nevertheless, the entropies of residues on βA in holo-HFABP are larger, which might be induced by the interactions between these residues and the ligand. The low conformational entropies of the residues on the bottom FABP indicate the structural rigidity of this region. It is unlikely to form an entrance to enable ligand binding or unbinding in the timescale of our simulation.

### 2.2. Intermediate States of Apo-FABP

The Markov state model (MSM) method was employed to determine the “invisible” intermediate states. Five intermediate states were characterized based on clustering on the tICA embedding space (Figure 3). The occupancies of the five intermediates were 32.8%, 19.4%, 18.7%, 15.5%, and 13.7%, respectively. The representative structures of these states overlapped with the experimental crystal structure (gray colored) are given in Figure 3. State A3 has the most stabilized secondary structures and the smallest RMSD from the experimental structure of apo-HFABP; therefore, A3 was treated as the “closed” state. It should be noted that the C-terminus of α2 in A2 is partially unfolded, but the Cα chemical shift values of conformations in A2 and A3 are almost identical to each other (Appendix A). The structural diversities between these intermediates are mainly located in the binding portal, including the α2 helix, the βC-βD loop, βE-βF loop, and βI-βJ loop, and the most flexible region was the α2 helix and the loop after it.

The convergence of the slowest relaxation time scales was tested to validate the Markovian assumption of transitions between the intermediates. The slowest relaxation time scale starts to level off at about 160 ns; therefore, a lag time was selected to calculate the net effective flux (Figure 4). The apparent free energy barriers between state pairs were calculated to estimate the transition kinetics. The apparent free energy barrier is related to the total net flow between two states, which is the number of transitions from the source to the sink per unit time. The lower the apparent free energy barrier between two states, the easier transitions occur between these states. The transition barriers between the “closed” state (A3) and other states are very low (Figure 3), e.g., a free energy barrier of only 2.5 kcal·mol^−1^ needs to be overcome from state A2 to state A3. Based on the apparent free energy barriers, the five intermediates can be further divided into two basins: one includes states A1 and A5, and the other includes A2, A3, and A4. The major structural difference between the conformations in the two basins is the unstructured level of the α2 helix. In the states A1 and A5, the α2 helix is more unstructured and results in a larger entrance portal, which indicates that the system needs to overcome a higher energy barrier to get more open structures for FABP.

The unfolding of the α2 helix was proposed as the critical step for the ligand binding of FABP. Although the intermediate states with unfolded α2 have not been directly observed in experiments, Yang’s group found via H–D exchange experiments that two or three states exist on the α2 region, and these states might play important roles in the ligand binding process. Based on stopped-flow experiments and NMR relaxation dispersion experiments, they proposed that the transition rate between the native “closed” state and the opened state is about 75,000 s^−1^ (transition time is about 13 μs). To investigate the kinetic transitions between the intermediate observed in this study and to find out whether the intermediates are related to the ligand binding process, we calculated the mean first-passage times (MFPT) between the intermediates of apo-HFABP (Table 1). The MFPT is the average first arrival time from a given source state to the given sink state and has been widely used in the study of the kinetics of protein folding and conformation exchanges. The MFPTs among the intermediates of apo-FABP range from several μs to more than 100 μs, which means that fast conformation exchanges happen between these states. Moreover, the transition times from the closed state (A3) to the other open states are in the order of dozens of μs, which is consistent with the time scale of the opening of apo-HFABP observed in experiments. Therefore, the intermediates found in our simulations might correspond to the states proposed by Xiao et al. [15], which are relevant to the binding process and are “invisible” to experiment technologies.

The unstructured α2 helix would form an entrance to allow the FA ligand to enter and therefore form an open gate for ligand binding (Appendix A). The side-chain solvent-accessible surface areas (SASAs) were calculated to characterize the contact probability of residues with outside ligands in different intermediates. Many residues are more exposed in the “open” states than in the native closed state, and most of them are located in the portal region (Appendix A). Interestingly, residue R31 interacts with residue D18 on α1 and is buried in the “closed” conformations; however, the residue stretched out to the solvent in the “open” states (Figure 5A). This result is consistent with the work by Young et al., in which they proposed that the positive charge residues on the portal region would attract the negatively charged terminus of FA and would be the driving force for binding [27]. In addition, the SASA values of A34 and F58 in the open states are larger than those in the “closed” state A3 (Figure 5B). In the native “closed” state, the hydrophobic residues V33 and A34 on the α2 helix and F58 on the βC-βD loop are in contact with each other, preventing outside ligands from entering the binding pocket (Figure 5C). On the other side, the unfolding of α2 in the open states would undermine the hydrophobic interactions and facilitate ligand binding (Figure 5D).

### 2.3. Intermediate States of Ligand-Binding HFABP

Six intermediate states of ligand-bound HFABP were characterized by MSM analysis (Figure 6). The largest state, H1, was characterized as the “closed” state since it adopts stable secondary structures and a similar overall structure to the experimental structure of holo-FABP (the Ca RMSD average over all conformations in H1 and the experimental structure is 1.62 Å). The population of the “closed” state of holo-FABP is larger than that of apo-FABP, which is consistent with the hypothesis of ligand binding shifting the conformational space of FABP to the closed form. Other than the native “closed” state, two kinds of intermediates were characterized based on the secondary and tertiary structural features. Like in apo-HFABP, the α2 helices in the first kind of states of holo-HFABP were partially unfolded (states H3 and H5). The other kind of state is related to the overall structure distortion (H2, H4, and H6), especially gap opening between the β-strands βD and βE.

The six intermediates of holo-FABP can be divided into two basins based on the apparent free energy barriers (Figure 6). The intermediates H3 and H5, with the unstructured α2 helix, are in the same basin with the native “closed” state H1. The states H2, H4, and H6, which have the incompact β-barrel structure, have larger free energy barriers than the “closed” state. The results indicate that unfolding of the α2 helix is much easier than the gap opening–closing transitions between the β-strands in the β-barrel.

There are many common structural properties in the apo-form and holo-form of HFABP. For example, flexible and unstructured α2 regions were observed in both states A1 and H3. The pairwise RMSDs between the representative structures of intermediates of apo- and holo-form HFABP are given in Appendix A. The low RMSD values between the representative structures (e.g., A1 and H3, A3 and H1) imply the similarity of these states. The clustering results of all conformations including apo- and holo-HFABP structures further demonstrate the overlap of the conformational spaces of the apo- and holo-HFABP (Appendix A). In the largest cluster, almost half of the conformations come from apo-HFABP (48%) and the other half come from holo-HFABP (52%). However, it is hard to make direct connections between the apo-states and holo-states based on the simulation data in this work. In simulations of the holo-system, the ligand basically stayed in the binding pocket; therefore, we do not have enough data to integrate the A states and H states due to the lack of transitions between these states.

The MFPTs between the intermediates of holo-FABP were calculated (Table 2). The transitions from the “closed” state H1 to the “open” states of holo-FABP are much slower than those for apo-FABP. The fastest transition leaving state H1 of holo-FABP is to the state H3, but the MFPT is about 7 times larger than the fastest transition in apo-FABP (i.e., from A3 to A1). This result indicates that ligand binding would not only shift the conformation population of the “closed” state but would also affect the free energy barriers between intermediates and the kinetic properties. There are two time scales from the “closed” H1 state to other states: The first one is from H1 to an α2 partially unfolded state, H3 or H5, and the MFPTs are less than 200 µs. The other is from H1 to H2, H4, or H6, whose βD and βE are separated, and the MFPTs are more than 600 μs.

Allosteric regulation, which happens at the allosteric sites, induces conformational and functional transitions of proteins [28]. The identification of allosteric sites would greatly promote the discovery of novel drugs [29,30]. However, the so-called hidden allosteric sites are invisible in experimental structures and are difficult to detect [31]. It is proposed that the hidden allosteric sites correspond to the ligand binding sites that emerged in the intermediates but are absent in the native structure. By analyzing the ligand–protein contact differences between the open intermediates and the crystal structure, many residues around the binding portal region have high contact probability with ligands in the open intermediates of HFABP, including L11 (13%), T119 (76%), H120 (31%), A123 (19%), and V124 (17%). These residues would contribute to the search for allosteric molecules to regulate the function of FABP.

### 2.4. Ligand Binding Model

By introducing a disulfide bond linking the α2 helix and the βE-βF loop, Yang et al. built a cap-closed FABP system. The cap-closed FABP still could bind oleic acids. Stopped-flow experiments indicated that there are at least two steps in the ligand binding process. The conformational exchanges between the “closed” and open states of FABP, which happen on a microsecond time scale, might be the rate-limiting step in ligand binding. Further NMR relaxation dispersion and hydrogen–deuterium experiments demonstrated the existence of many “invisible” minor intermediates, and the ligand-binding-relevant open state has structures with local unfolding of the α2 helix [15]

The intermediates revealed by our extended-time simulations are in good agreement with the experimental data. For ligand-unbound FABP, the α2 helices of the intermediates other than the native “closed” state are partially unfolded, which is consistent with the H/D exchange experiment data. On the other hand, gap-open states were not observed in our simulations. Besides this, the transition times between the “closed” state and the open states with partially unfolded α2 helices revealed in our study meet with experimental estimations. Ligand binding would slow down the transitions between the native “closed” state and the open intermediates; however, the MFPTs of these states are still on a microsecond time scale. Despite the lack of direct evidence, Yang et al. assumed that the open-to-closed transitions of ligand-bound human intestine FABP are related to the slow step of the ligand association process, which is in the millisecond time scale. The inconsistency might be caused by the different types of FABPs studied in the experiment and simulations. The HFABP used in our simulation has 10-fold higher affinity than the IFABP used in the experiments, which indicates that HFABP has low free energy barriers in the ligand binding process and faster transition rates between the intermediates.

Our simulations provided atomic-level structures of the open intermediate states in the ligand binding process. A hypothetical model was proposed to describe the accessing and releasing process of fatty acid with HFABP (Figure 7). Based on our results, the transitions from the native “closed” state of apo-HFABP to the open state are very quick (less than 100 μs), and the open conformations with unstructured α2 helices have a larger binding cavity which facilitates the binding of fatty acids. The positively charged residues on the portal region (especially the residue R31) prefer to stretch out to the solvent, which provides the driving force to bind the negative terminus of the ligands (Step B). Therefore, the binding of ligands to HFABP might obey the conformational selection mechanism. Ligand binding would stabilize the native “closed” structure, which would allow the transportation of the ligand. There are two kinds of ligand-bound “open” structures, which indicates that there are two ligand releasing pathways, i.e., the α2 helix partially unfolded pathway (D1 and D2 steps) and the βD-βE gate open pathway (E1 and E2 steps). It should be noted that our model only gives possible structure dynamics of FABP in the FA binding and unbinding process; further elaborate experiments and exhaustive computation studies are still required to uncover the detailed binding/unbinding mechanisms.

## 3. Materials and Methods

### 3.1. Molecular Dynamics (MD) Simulation 

All-atom molecular dynamics (MD) simulations were performed on apo-HFABP and holo-HFABP, respectively. The X-ray experiment structures were used as the initial structure for the MD simulations of apo-HFABP and holo-HFABP (Figure 1). The experimental X-ray structures of apo-HFABP and holo-HFABP are almost the same; the root-mean-square deviation (RMSD) between the two experimental structures is 0.84 Å. The crystallographic water molecules in the initial structures were removed. The TIP3P water box was built to solvate the protein, and the solute atoms were at least 12.5 Å away from the boundary of the rectangular box. Sodium ions were added as the counter ions to neutralize the net charge of each system. The AMBER99SB-ILDN force field [32] and the general amber force field (gaff) [33] were used for the protein and the ligand. The whole system was composed of protein, ligand, counter ions, and water including 28,613 atoms with a box of 72 × 66 × 75 Å^3^ and 27,499 atoms with a box of 70 × 66 × 75 Å^3^ for the simulations of the apo-FABP and holo-FABP, respectively. The systems were minimized by a 2000-step steepest decent method and a 2000-step conjugate gradient method, whereas the protein and ligand were constrained with a harmonic force of 100 kcal mol^−1^ Å^−2^. Then, the systems were heated up from 0 K to 300 K gradually and equilibrated without restraint with the volume held constant over 100 ps. Finally, the systems were equilibrated without restraint for 110 ps with a Langevin thermostat in an NPT (P = 1 atm and T = 300 K) ensemble. During MD simulations, the particle mesh Ewald (PME) method [34] was adopted to calculate the long-range electrostatic interactions. The non-bonded cutoff was set to 9 Å, and all bonds involving hydrogen atoms were fixed to their equilibrium values using the SHAKE algorithm [35]. Finally, ten MD simulation production runs with a cumulative simulation time of 60 µs were collected by cuda-version Amber 14 [36]: four 6 µs trajectories for apo-FABP and six 6 µs trajectories for holo-FABP.

### 3.2. Markov State Model Analysis

To investigate the thermodynamic and kinetic properties of apo-FABP and holo-FABP, a Markov state model (MSM) was built using the program MSMBuilder3.6 [37] as follows: Firstly, the conformations of the MD trajectories were transformed into the dihedral angle matrix. Then, using time-lagged independent component analysis (tICA) [38], the conformational space was projected to a four-dimensional space. Based on the tICA embedding, the conformations were clustered into 500 states (microstates) by using the *k*-means clustering algorithm. The conformations in the same cluster are geometrically similar and interconvert rapidly. The transition matrix between the microstates was then constructed using maximum likelihood and Bayesian estimation at a proper lag time (40 ns for apo-HFABP and 30 ns for holo-HFABP). The implied time scales converged and the transitions between the microstates become the Markovian process at the lag time we chose (Appendix A). The microstates were lumped into macrostates by Perron cluster cluster analysis (PCCA+) [39] based on the kinetic similarities of the microstates. MSMs with five macrostates (intermediate states) and six macrostates were built for apo-HFABP and holo-HFABP, respectively.

### 3.3. Apparent Free Energy Barrier

The apparent free energy barriers between the intermediates were calculated by following the method used in Ref [40]. First of all, the net flux (*f_ij_^+^*) between microstate *i* in a given source macrostate *s* and microstate *j* in a given sink microstate *t* were calculated as follows:(1)$$f_{ij}=\pi_{i}q_{i}^{-}T_{ij}q_{j}^{+}$$
where *i* and *j* are nodes (or microstates); *T_ij_* is the transition probability between *i* and *j*; $q_{j}^{+}$ and $q_{i}^{-}$ are the forward committor probability and the backward committor probability, respectively; and $\pi_{i}$ is the Boltzmann distribution of the microstates. The total net flow leaving the source (*S*) and entering the sink (*T*) was then calculated by summing for all microstates *i* in *S* and *j* in the sink *T*:(2)$$F_{ST}=\sum_{i\in s,j\notin s}f_{ij}^{+}$$

The physical meaning of *F_ST_* is the effective transitions from the source (reactant) to the sink (product) during the lag time *τ*. The apparent free energy barrier between the source and sink (*F_ST_^*^*) is related to *F_ST_* by
(3)$$F_{ST}^{*}(TPT)=-k_{B}T\ln\left(\frac{F_{st}}{\tau }\frac{h}{k_{B}T}N_{total}\right)$$
where *h* is Planck’s constant, *T = 300 K*, and the lag time *τ* is 120 ns for apo-HFABP and 160 ns for holo-HFABP. *N_total_* is the total number of conformations. Then, the pairwise free energy barriers were used as the edge weights to build the transition networks between the macrostates.

### 3.4. Mean First Passage Time

The mean first passage time (MFPT) is defined as the averaged first arrival time from the source state to the sink state [41]. MFPTs can therefore describe the kinetics information of the system. The MFPT from the microstate *i* in the source macrostate *S* to the final sink state *T* can be calculated as
(4)$$MFPT_{iT}=\sum_{k\in S}T{\left(\tau \right)}_{ik}(\tau +MFPT_{kT})$$
where T(τ) is the transition probability matrix and *τ* is the transition lag time of the system; here, *τ* is 160 ns for apo-HFABP and 120 ns for holo-HFABP. The MFPT between the source state *S* and the sink state *T* can be calculated by
(5)$$MFPT_{ST}=\sum_{i\in S}p_{i}*MFPT_{iT}$$
where $p_{i}$ is the normalized population of the microstate *i* in the source state *S*.

## 4. Conclusions

Atomic MD simulations covering a total of 60 μs were performed on ligand-free HFABP and ligand-bound HFABP to characterize the “invisible” intermediates near the native structures. All secondary structure elements except the α2 helix were well preserved in the simulations of both apo- and holo-HFABP. The transition rates between these intermediate states and the native “closed” state are in good agreement with the experimental measurements, which indicates that the intermediate states are binding-relevant states. The orientation change of R31 on the α2 helix might provide the driving force to attack the ligand to bind with FABP. Ligand binding would stabilize the “closed” structure of FABP, which would facilitate the transportation of the fatty acids. There are two kinds of ligand-bound “open” structures, which might relate to different ligand releasing pathways, i.e., the α2 helix partially unfolded pathway and the βD-βE gate open pathway.

## Figures and Tables

**Figure 1 ijms-20-03476-f001:**
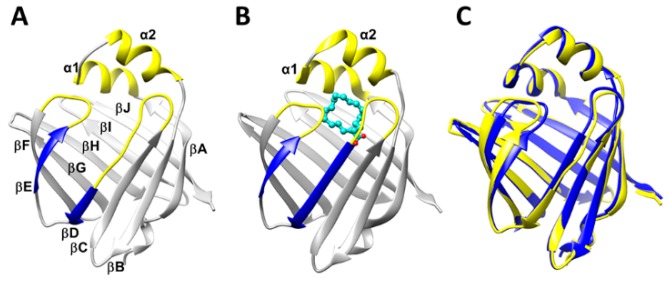
Structure of heart fatty acid binding protein (HFABP): (**A**) The 3D structure of ligand-free HFABP (apo-HFABP, PDB ID: 3RSW). The dynamic portal region is colored in yellow, and the βD and βE are colored in blue. (**B**) The 3D structure of ligand-binding HFABP (holo-HFABP, PDB ID: 3WXQ). The stearic acid is shown in ball-and-stick form. (**C**) The overlap of the experimental structures of apo- and holo-HFABP.

**Figure 2 ijms-20-03476-f002:**
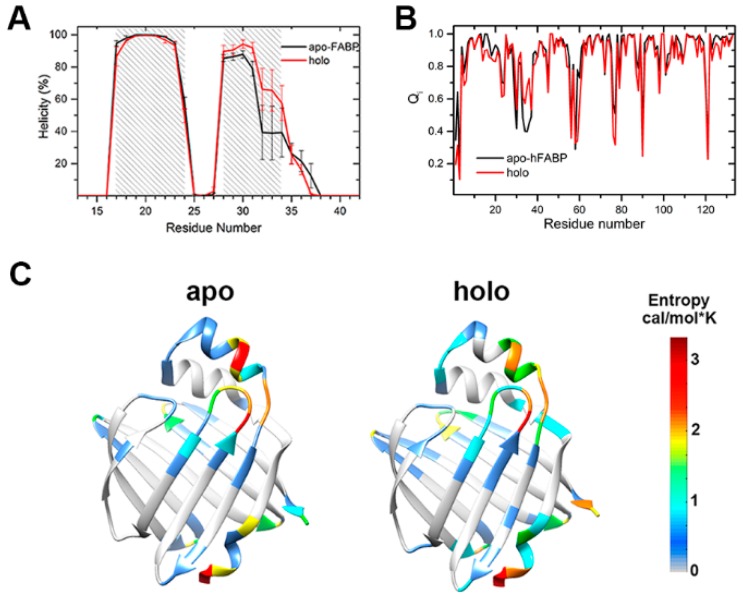
Structural properties of hFABP: (**A**) The helicity of the α1 and α2 helices; the error bars represent the mean square error in different trajectories. (**B**) The fraction of residue native contacts. (**C**) The backbone conformational entropy of each residue. A color map of the entropies is also given. The residues colored in red have large conformational entropies, and blue colors correspond to small entropies. The residues with entropy equal to 0 are colored in gray.

**Figure 3 ijms-20-03476-f003:**
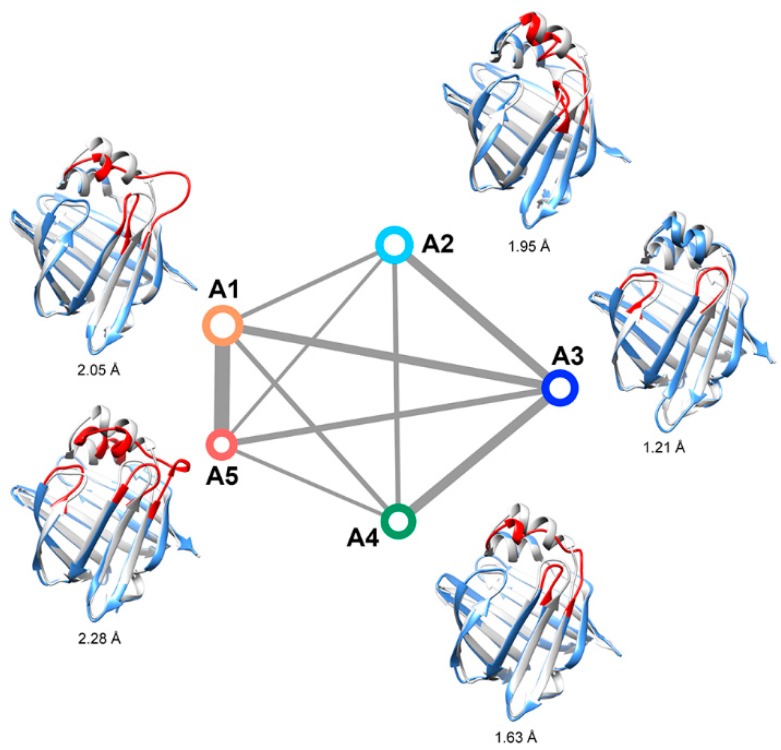
Kinetic network of the intermediates of apo-FABP. The size of each node is proportional to the population of the corresponding intermediate state. The width of each edge is proportional to the apparent free energy barrier between the two nodes. The root-mean-square deviations (RMSDs) between the representative structure (yellow) of the states and the crystal structure (gray) are given, and the regions with large structural differences are shown in red.

**Figure 4 ijms-20-03476-f004:**
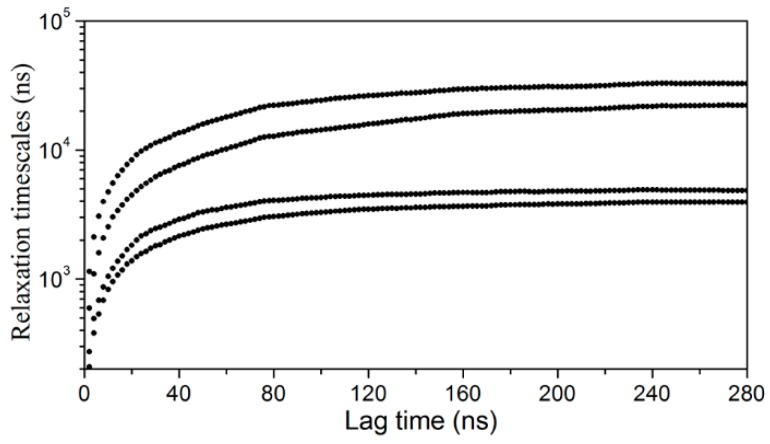
Implied time scale plot as a function of lag time for a five-macrostate model.

**Figure 5 ijms-20-03476-f005:**
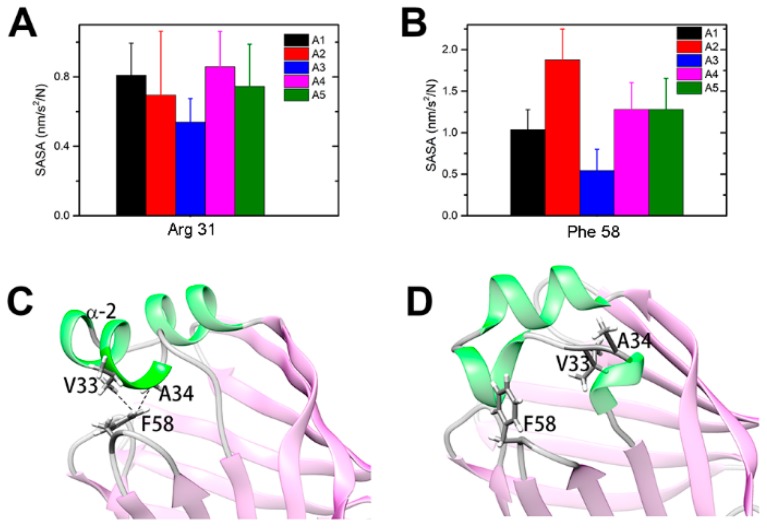
The solvent-accessible surface area (SASA) and contact analysis of apo-HFABP: (**A**) SASA of Arg31 in different intermediates of apo-HFABP; (**B**) SASA of Phe58 in different intermediate states; (**C**) The hydrophobic interactions between residue F58 and V33/A34 in the “closed” state A3; (**D**) The interactions between F58 and V33/A34 were broken in the open intermediate states. The SASAs were calculated by the gmx-sasa module implemented in Gromacs.

**Figure 6 ijms-20-03476-f006:**
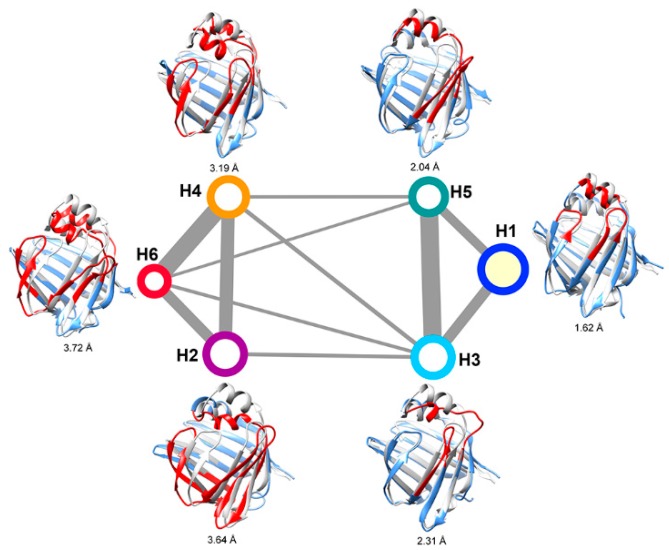
Kinetic network of holo-FABP. The sizes of nodes and widths of edges are defined the same as those in Figure 3.

**Figure 7 ijms-20-03476-f007:**
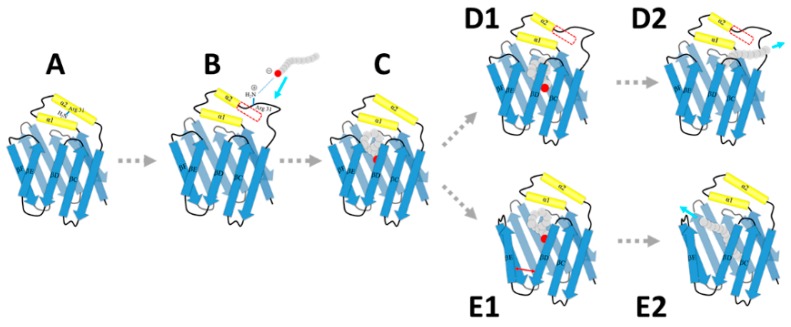
A schematic model of fatty acid associating and disassociating with heart FABP. The dashed arrows indicate hypothetical transitions between the intermediates. The regions in the red frames are unfolded in the corresponding intermediates. The cyan arrows are corresponding to the moving directions of the ligands. The red arrow represents the movement of two β-strands.

**Table 1 ijms-20-03476-t001:** The mean first-passage times (MFPTs) between the intermediate states of apo-HFABP (unit: µs).

	A1	A2	A3	A4	A5
A1	0	124.87	63.66	124.81	25.84
A2	36.11	0	13.54	76.22	60.47
A3	22.58	61.22	0	62.91	46.94
A4	24.82	64.98	4.0	0	49.08
A5	2.51	125.91	64.69	125.75	0

**Table 2 ijms-20-03476-t002:** The MFPTs between the intermediate states of holo-HFABP (unit: µs).

	H1	H2	H3	H4	H5	H6
H1	0	639.15	144.61	616.00	199.05	671.48
H2	74.08	0	41.95	48.47	92.98	103.68
H3	32.12	494.54	0	471.39	54.44	526.87
H4	73.52	71.07	41.40	0	92.07	55.67
H5	33.48	492.49	1.36	468.98	0	524.47
H6	74.54	71.81	42.42	1.20	93.10	0

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
