# Peer review of "The Observation of Ligand-Binding-Relevant Open States of Fatty Acid Binding Protein by Molecular Dynamics Simulations and a Markov State Model"

_ijms, 2019, doi:10.3390/ijms20143476_

Reviewer 1 Report

In the present manuscript, Guo and coworkers perform detailed computational studies of ligand binding to fatty acid binding protein (FABP), through extensive simulations and Markov state analysis. They identified intermediate states for both ligand-unbound and -bound FABP, and explored the transition rate between the microstates.

The calculations are solid and the manuscript is well written, and I struggled to find anything to criticize. The one comment I have is that the authors could discuss extensively in the manuscript whether the existence of hidden allosteric sites in the intermediate states, because hidden allosteric sites that are invisible in the crystal structures are important drug targets (PMID: 30817889, 30688063, 29030241, 29457894).

If the authors address this issue, I can only enthusiastically recommend this manuscript for publication in IJMS.

Author Response

We wish to thank the referee for a careful reading of our manuscript. The referee provided us valuable comments and advices. We revised our manuscripts accordingly. In the following your comments are in blue italics and our response is in regular font.

 In the present manuscript, Guo and coworkers perform detailed computational studies of ligand binding to fatty acid binding protein (FABP), through extensive simulations and Markov state analysis. They identified intermediate states for both ligand-unbound and -bound FABP, and explored the transition rate between the microstates.

The calculations are solid and the manuscript is well written, and I struggled to find anything to criticize. The one comment I have is that the authors could discuss extensively in the manuscript whether the existence of hidden allosteric sites in the intermediate states, because hidden allosteric sites that are invisible in the crystal structures are important drug targets (PMID: 30817889, 30688063, 29030241, 29457894).

If the authors address this issue, I can only enthusiastically recommend this manuscript for publication in IJMS.

Reply: We would like to thank the reviewer for the valuable suggestions. Allosteric regulation, which happened on the allosteric sites, would induce the conformational and functional transitions of proteins. The identification of the allosteric sites would greatly promote the discovery of novel drugs. However, the so called hidden allosteric sites are invisible in the experimental structures. By analyzing the ligand-protein contact differences between the open intermediates and the crystal structure, we found many interesting residues around the binding portal region might be the hidden allosteric sites of HFABP. The following discussion were added to the text:

“Allosteric regulation, which happened on the allosteric sites, induce the conformational and functional transitions of proteins.[38] The identification of the allosteric sites would greatly promote the discovery of novel drugs.[39, 40] However, the so called hidden allosteric sites are invisible in the experimental structures and are difficult to be detected.[41] It is proposed that the hidden allosteric sites are corresponding to the ligand binding sites emerged in the intermediates which are absent in the native structure. By analyzing the ligand-protein contact differences between the open intermediates and the crystal structure, many residues around the binding portal region have high contact probability with the ligands in the open intermediates of HFABP, including L11 (13%), T119 (76%), H120 (31%), A123 (19%) and V124 (17%). These residues would contribute to the search of allosteric molecules to regulate the function of FABP.”

Reviewer 2 Report

The paper sheed light in the binding mechanism of fatty acid with FABP to understand the FABP functions.all-atom molecular dynamics (MD) simulations were performed to characterize the  structure properties of native-like intermediate states that can be considered relavant for the bindings.

The paper is well written and perfomerd interesting simulations. Thare are only minor changes required.

Pag.3 line 101: the RMSD reported is very low 0.84 angstrom. it referes to the protein backbone? Could you superimepose the different domanis in order to put in evidence where are the major changes between the two structres?

- in the same section line 108: remove "hybrid" and specify the exact numer of steps used of each minimization algorithm

-pag 3 line 117: check the time of MD simulation

-overall check the used terminology as "binding relevant" : is not clear. it must be put in a more appropiate English

Author Response

We wish to thank the referee for a careful reading of our manuscript. The referee provided us valuable comments and advices. We revised our manuscripts accordingly. In the following your comments are in blue italics and our response is in regular font.

 The paper sheed light in the binding mechanism of fatty acid with FABP to understand the FABP functions. all-atom molecular dynamics (MD) simulations were performed to characterize the structure properties of native-like intermediate states that can be considered relavant for the bindings.

The paper is well written and perfomerd interesting simulations. There are only minor changes required.

Pag.3 line 101: the RMSD reported is very low 0.84 angstrom. it referes to the protein backbone? Could you superimepose the different domanis in order to put in evidence where are the major changes between the two structres?

Reply: We would like to thank the reviewer for the valuable advices. The experimental structures of apo- and holo-HFABP are almost identical to each other, the Ca RMSD between the closed form of these two structures is 0.84 angstrom. The superposition of the two experiment structures was added to Figure 1. The main structures of apo- and holo-HFABP are identical to each other, while some subtle structure changes on the loop regions.

- in the same section line 108: remove "hybrid" and specify the exact numer of steps used of each minimization algorithm

Reply: We thank the reviewer for the suggestions. We changed the sentences to “The systems were minimized by 2000-step steepest decent method and 2000-step conjugate gradient method, whereas the protein and ligand were constrained with a harmonic force of 100 kcal mol-1 Å-2.”

-pag 3 line 117: check the time of MD simulation

Reply: In this study, we performed 10 six-ms trajectories. The total simulation time is 60 ms.

-overall check the used terminology as "binding relevant" : is not clear. it must be put in a more appropiate English

Reply: We thank the reviewer for the valuable suggestion. The concept of “binding relevant” intermediates is following the definition in the Xiao et al.’s paper (Angew. Chem. Int. Edit., 2016, 55:6869), which means the exchanges rate between these states to the closed native state agree with the experiment determined maximal ligand association rate. We added the explanation of “binding relevant” to the text in its first appearance: “The binding relevant intermediates, which are functionally important in the ligand binding process and defined by the experiment determined ligand association rate with the native closed state, are “invisible” and hard to be determined by experimental technologies.”

 Reviewer 3 Report

      In my opinion the biological significance is great, and this is an interesting paper, will be a good contribution for this kind of article in the International Journal of Molecular Sciences.

Author Response

 Reply: We would like to thank the reviewer for the supporting.

Reviewer 4 Report

The manuscript by Yue Guo and colleagues describes the attempt to probe the mechanism of ligand binding by the the heart fatty acid binding protein using molecular modeling. The problem is actual, the choice of method is reasonable and prospective since experimental study of such fast transitions is difficult. However, some points of the analysis and conclusion needs to be revised.
Main question: the logics from modeling of two states (apo and holo) to the transition model is poor or unclear.
- First, Authors compare different states of apoform and holoform in details inside the groups, but do not compare the structures of apoprotein (A) states with structures of holoenzyme (H) states. Are there any H states which are structurally similar to any A state? Maybe, H3, or probably a combined A-H clustering can help? Such intersection of A and H states could indicate an actual intermediate form.
- What about ligand behavior during holoform simulations? Did it release?
- Are the transitions analyzed in the manuscript (especially formation of open states H2 and H6) reversible?
Summarizing, authors studied fluctuations of to forms of the FABP (apo and holo), but not the conversion of them to each other. To conclude about the binding and release mechanism, some interconnections between two forms should be found. It may be the same conformation or modeling of the ligand release (for example by steered MD), or maybe modeling of ligand binding after placing them close to the gate in open state of the apoform.
Minor points:
- l.242-243 - "The results indicate that the intermediates found in our simulations are ligand binding relevant" – I am afraid that the fact that time scale of the processes in simulation and in experiments is not enough to claim that it it the same process. This should be rephrased.
- fig.3 and fig.6 - initial state (gray) is too pale, difficult to see it.
- please comment (maybe in introduction) why it is impossible that the ligand molecule penetrates into the cavity from the"bottom" side.
- I suggest to show the open gates in surface or sticks representation for better observation that the gate is actually open.
- l.59 – abbreviation "H-D experiments" should be described
- l.73 - toad live FABP

Author Response

We wish to thank the referee for a careful reading of our manuscript. The referee provided us valuable comments and advices. We revised our manuscripts accordingly. In the following your comments are in blue italics and our response is in regular font.

 The manuscript by Yue Guo and colleagues describes the attempt to probe the mechanism of ligand binding by the the heart fatty acid binding protein using molecular modeling. The problem is actual, the choice of method is reasonable and prospective since experimental study of such fast transitions is difficult. However, some points of the analysis and conclusion needs to be revised.
Main question: the logics from modeling of two states (apo and holo) to the transition model is poor or unclear.
- First, Authors compare different states of apo-form and holo-form in details inside the groups, but do not compare the structures of apoprotein (A) states with structures of holoenzyme (H) states. Are there any H states which are structurally similar to any A state? Maybe, H3, or probably a combined A-H clustering can help? Such intersection of A and H states could indicate an actual intermediate form.

 Reply: We would like to thank the reviewer for the valuable comments. There are many common structure properties in the apo-form and holo-form HFABP. For example, the flexible and unstructured a2 regions were observed in both states A1 and H3. The combined A-H clustering demonstrated the overlap of the conformational space of the apo- and holo-HFABP: in the largest cluster, almost half of the conformations come from the apo-HFABP (48%) and the other half come from the holo-HFABP (52%). However, it is hard to make the direct connections between the apo-states and holo-states based on the simulation data in this work. In the simulations of the holo-system, the ligand was basically stay in the binding pocket, therefore, we don’t have enough data to integrate the A states and H states due to the lack of the transitions between these states.

- What about ligand behavior during holoform simulations? Did it release?

 Reply: The ligand stays inside the binding pocket during most of our simulations. Actually, the releasing of the ligand only observed in one of the six holo-HFABP trajectories, and the ligand re-enter the protein in a short time after the releasing. The releasing conformations only account for 0.6% of the total conformations. Removing these conformations doesn’t change the structure properties, and have very limit influence on the transition matrix and therefore would not affect the conclusions obtained by the MSM analysis.
- Are the transitions analyzed in the manuscript (especially formation of open states H2 and H6) reversible?

Reply: Yes, the transitions are reversible. The transitions from closed states to the open states, as well as the transitions from open states to the closed states were frequently observed in our simulations. The convergence test by MSM analysis demonstrated that the transitions between different states obey the detailed balance. The MFPTs analysis therefore can give the transition rate from state A to B and the reverse process, i.e. B to A.

Summarizing, authors studied fluctuations of to forms of the FABP (apo and holo), but not the conversion of them to each other. To conclude about the binding and release mechanism, some interconnections between two forms should be found. It may be the same conformation or modeling of the ligand release (for example by steered MD), or maybe modeling of ligand binding after placing them close to the gate in open state of the apoform.

 Reply: We would like to thank the reviewer for the valuable comments. In this paper, we mainly focused on the structures of near native intermediates of HFABP and the structure differences between the apo- and holo-HFABP. Based on the intermediates found in this study and the binding steps observed by Xiao et al.’s experiments (Angew. Chem. Int. Edit., 2016, 55:6869), we proposed a hypothetical model to describe the possible structural dynamics in the binding and disassociation process, which would help to understand the binding mechanism. However, it is impossible to uncover the detailed association and disassociation mechanism based on the current tradition MD simulation technology. We added the following sentences to the page 10: “It should be noticed that our model only give the possible structure dynamics of FABP in the FA binding and unbinding process, the further elaborate experiments and exhaustive computation studies are still required to uncover the detailed binding/unbinding mechanisms.”. Besides, we changed the model accordingly: the arrow between the intermediates were replaced by the dashed line.

We also thank the reviewer for providing the useful suggestions, actually we are working on using the enhanced sampling technologies to study the binding mechanism, hopefully we can give a more detailed picture to describe the binding/unbinding of FA in the near future.
Minor points:
- l.242-243 - "The results indicate that the intermediates found in our simulations are ligand binding relevant" – I am afraid that the fact that time scale of the processes in simulation and in experiments is not enough to claim that it it the same process. This should be rephrased.

 Reply: We thank the reviewer for the valuable suggestions. We changed it to: “Therefore, the intermediates found in our simulations might corresponding to the states proposed by Yang et al., which are relevant to the binding process and are “invisible” to the experiment technologies.”

- fig.3 and fig.6 - initial state (gray) is too pale, difficult to see it.

 Reply: Thank the reviewer for the valuable advice. We changed the color in the figures.

- please comment (maybe in introduction) why it is impossible that the ligand molecule penetrates into the cavity from the “bottom” side.

 Reply: The following comments were added to the last paragraph of page 5: “The low conformational entropies of the residues on the bottom FABP indicating the structure rigidity of this region. It is unlikely to form an entrance to enable the ligand binding or unbinding in the timescale of our simulation.”

- I suggest to show the open gates in surface or sticks representation for better observation that the gate is actually open.

 Reply: We thank the reviewer for the suggestion. The surface representation of the examples of open states was given in Figure S5.

- l.59 – abbreviation “H-D experiments” should be described

 Reply: Thank the reviewer for the advice. We replaced the abbreviation “H-D experiments” by “hydrogen-deuterium (H-D) exchange experiments” in its first appearance.

- l.73 - toad live FABP?

 Reply: Yes, the live FABP studied in Ref.21 came from Toad. Toad is a kind of frog.

Round  2

Reviewer 4 Report

Authors’ reply, “For example, the flexible and unstructured a2 regions were observed in both states A1 and H3. The combined A-H clustering demonstrated the overlap of the conformational space of the apo- and holo-HFABP: in the largest cluster, almost half of the conformations come from the apo-HFABP (48%) and the other half come from the holo-HFABP (52%).”

I recommend to include a comparison of A1 and H3 states (and others) into the text, not only into the reply to reviewer. The same for combined A-H clustering: show the structure of at least the largest cluster and discuss the “intermediate” states and the transition model from this point of view.

Also, a comprehensive proofreading is required. For example, toad liver FABP was studied in the cited paper, not the live of toad or “live FABP” from toad.

Author Response

Authors’ reply, “For example, the flexible and unstructured a2 regions were observed in both states A1 and H3. The combined A-H clustering demonstrated the overlap of the conformational space of the apo- and holo-HFABP: in the largest cluster, almost half of the conformations come from the apo-HFABP (48%) and the other half come from the holo-HFABP (52%).”

 - I recommend to include a comparison of A1 and H3 states (and others) into the text, not only into the reply to reviewer. The same for combined A-H clustering: show the structure of at least the largest cluster and discuss the “intermediate” states and the transition model from this point of view.

 Reply: We would like to thank the reviewer for the suggestions. We did the pairwise RMSD between the representative structures of the intermediates in apo- and holo-HFABP (Table S1). Also, we added the analysis of combined A-H clustering into Table S2. The following discussion were added to the text:

“There are many common structure properties in the apo-form and holo-form HFABP. For example, the flexible and unstructured a2 regions were observed in both states A1 and H3. The pairwise RMSDs between the representative structures of intermediates of apo- and holo-form HFABP were given in Table S1. The low RMSD values between the representative structures (e.g. A1 and H3, A3 and H1) implied the similarity of these states. The clustering results of all conformations including apo- and holo-HFABP structures further demonstrated the overlap of the conformational space of the apo- and holo-HFABP (Table S2). In the largest cluster, almost half of the conformations come from the apo-HFABP (48%) and the other half come from the holo-HFABP (52%). However, it is hard to make the direct connections between the apo-states and holo-states based on the simulation data in this work. In the simulations of the holo-system, the ligand was basically stay in the binding pocket, therefore, we don’t have enough data to integrate the A states and H states due to the lack of the transitions between these states.”

 - Also, a comprehensive proofreading is required. For example, toad liver FABP was studied in the cited paper, not the live of toad or “live FABP” from toad.

 Reply: We thank the reviewer for the valuable advices. We carefully checked the spelling in the manuscript. We changed the typo to“the Toad liver FABP”.